# Data Poisoning Won't Save You From Facial Recognition

**Evani Radiya-Dixit**
Stanford University

**Sanghyun Hong**
Oregon State University

**Nicholas Carlini**
Google

**Florian Tramèr**
Stanford University, Google

## Abstract

Data poisoning has been proposed as a compelling defense against facial recognition models trained on Web-scraped pictures. Users can perturb images they post online, so that models will misclassify future (unperturbed) pictures.

We demonstrate that this strategy provides a false sense of security, as it ignores an inherent asymmetry between the parties: users' pictures are perturbed *once and for all* before being published (at which point they are scraped) and must thereafter fool *all future models*—including models trained adaptively against the users' past attacks, or models that use technologies discovered after the attack.

We evaluate two systems for poisoning attacks against large-scale facial recognition, *Fawkes* (500,000+ downloads) and *LowKey*. We demonstrate how an "oblivious" model trainer can simply *wait* for future developments in computer vision to nullify the protection of pictures collected in the past. We further show that an adversary with black-box access to the attack can (i) train a robust model that resists the perturbations of collected pictures and (ii) detect poisoned pictures uploaded online.

We caution that facial recognition poisoning will not admit an "arms race" between attackers and defenders. Once perturbed pictures are scraped, the attack cannot be changed so any *future* successful defense irrevocably undermines users' privacy.

## 1 Introduction

Facial recognition systems pose a serious threat to individual privacy. Various companies routinely scrape the Web for users' pictures to train large-scale facial recognition systems (Hill, 2020a; Harwell, 2021), and then make these systems available to law enforcement agencies (Lipton, 2020) or private individuals (Harwell, 2021; Mozur & Krolik, 2019; Wong, 2019).

A growing body of work develops tools to allow users to fight back, using techniques from *adversarial machine learning* (Sharif et al., 2016; Oh et al., 2017; Thys et al., 2019; Kulynych et al., 2020; Shan et al., 2020; Evtimov et al., 2020; Gao et al., 2020; Xu et al., 2020; Yang et al., 2020; Komkov & Petiushko, 2021; Cherepanova et al., 2021a; Rajabi et al., 2021; Browne et al., 2020).

One approach taken by these tools lets users perturb any picture before they post it online, so that facial recognition models that train on these pictures will become *poisoned*. The objective is that when an *unperturbed* image is fed into the poisoned model (e.g., a photo taken by a stalker, a security camera, or the police), the model misidentifies the user. This approach was popularized by *Fawkes* (Shan et al., 2020), an academic image-poisoning system with 500,000+ downloads and covered by the New York Times (Hill, 2020b), that promises "strong protection against unauthorized [facial recognition] models". Following Fawkes' success, similar systems have been proposed by academic (Cherepanova et al., 2021a; Evtimov et al., 2020) and commercial (Vincent, 2021) parties.

This paper shows that these systems (and, in fact, any poisoning strategy) **cannot protect users' privacy**. Worse, we argue that these systems offer a false sense of security. There exists a class of privacy-conscious users who might have otherwise never uploaded their photos to the internet; however who now might do so, under the false belief that data poisoning will protect their privacy. These users are now *less private* than they were before. Figure 1 shows an overview of our results.

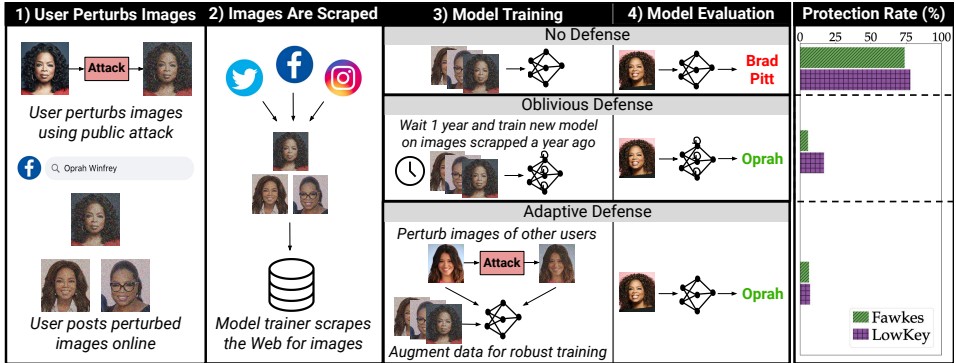

Figure 1: **Attacks and defenses for facial recognition poisoning.** (1) Users perturb their pictures before posting them online. (2) A model trainer continuously scrapes the Web for pictures. (3-4) The model trainer builds a model from collected pictures and evaluates it on *unperturbed* pictures. With no defense strategy, the poisoned model fails to recognize users whose online pictures were perturbed. An "oblivious" model trainer can wait until a better facial recognition model is discovered and retroactively train it on past pictures to resist poisoning. An adaptive model trainer with black-box access to the attack employed by users can immediately train a robust model that resists poisoning. We show the effectiveness of these defenses against the *Fawkes* and *LowKey* poisoning attacks.

The reason these systems are not currently private, and can never be private, comes down to a fundamental asymmetry between Web users and the trainers of facial recognition models. Once a user commits to an attack and uploads a perturbed picture that gets scraped, *this perturbation can no longer be changed*. The model trainer, who acts second, *then* gets to choose their training strategy. As prior work lacks a formal security setup, we begin by defining a security game to capture this *dynamic* nature of poisoning attacks.

We then introduce two powerful defense strategies that completely break two state-of-the-art poisoning attacks, Fawkes (Shan et al., 2020) and LowKey (Cherepanova et al., 2021a). In the first strategy, we *adapt* the facial recognition training to work in the presence of poisoned images. Because image-perturbation systems are made *publicly accessible* to cater to a large user base (Shan et al., 2021; Cherepanova et al., 2021b), we must assume facial recognition trainers are aware of these attack techniques. Our adaptive models fully circumvent poisoning with just *black-box* access to the attack.

Worse, we find there exists an even simpler defensive strategy: model trainers can *just wait* for better facial recognition systems, which are no longer vulnerable to these particular poisoning attacks. That is, because existing poisoning attacks were only designed to prevent *current* face recognition tools from working, there is no reason to believe that *future* tools will be poisoned as well. Indeed, we show that the state-of-the-art poisoning attacks are **already broken** by new training techniques that appeared less than a year later. For example, Fawkes (released in July 2020) is ineffective if the model trainer switches to a MagFace model (Meng et al., 2021) (released in March 2021), and LowKey (released January 2021) is ineffective against a facial recognition model obtained by finetuning OpenAI's CLIP model (Radford et al., 2021) (also released January 2021).

We argue that poisoning attacks against facial recognition will *not* lead to an "arms race", where new attacks can continuously counteract new defenses. Since the perturbation applied to a picture cannot be changed once the picture is scraped, a successful poisoning attack has to remain effective against *all* future models, even models trained adaptively against the attack, or models that use new techniques discovered only after the attack. In light of this, we argue that users' only hope is a push for legislation that restricts the use of privacy-invasive facial recognition systems (Singer, 2018; Weise & Singer, 2020; Winder, 2020).

## 2 DATA POISONING FOR FACIAL RECOGNITION

### 2.1 THREAT MODEL

We consider a setting where a *user* uploads pictures of themselves to an online service such as a social media platform. The user attempts to protect their pictures by adding perturbations that should

be almost imperceptible to other people (Szegedy et al., 2013). The user's goal is that a model trained on their perturbed pictures will achieve low accuracy when classifying *unperturbed* pictures of the user (Shan et al., 2020; Cherepanova et al., 2021a; Yang et al., 2020; Evtimov et al., 2020).

A second party, the *model trainer*, scrapes the Web for pictures to train a large-scale facial recognition model (capable of identifying a large number of users). We assume that the data scraped by the trainer is *labeled*, i.e., all (possibly perturbed) images collected of a user can be assigned to the user's identity. The trainer's goal is to build a model that correctly recognizes users in future images. The trainer is *active*, i.e., they continuously scrape new uploaded pictures at regular intervals.

This setting corresponds to that of *training-only clean-label* poisoning attacks (Shan et al., 2020; Cherepanova et al., 2021a; Goldblum et al., 2020; Evtimov et al., 2020). Keeping with the terminology of the data poisoning literature (Goldblum et al., 2020), we refer to the user as the *attacker* and the trainer as the *defender* (even though it is the trainer that aims to breach the user's privacy!).

## 2.2 Poisoning Attack Games

We present a standard security game for training-only clean-label poisoning attacks in Figure 8a. We argue that this game fails to properly capture the threat model of our facial recognition scenario.

In this game, the attacker first samples training data $\mathbf{X}, \mathbf{Y}$ from a distribution $\mathbb{D}$. The attacker then applies an attack to get the perturbed data $\mathbf{X}_{\mathrm{adv}}$. The defender gets the perturbed labeled data $(\mathbf{X}_{\mathrm{adv}}, \mathbf{Y})$ and trains a model $f$. The model $f$ is evaluated on *unperturbed* inputs $x$ from the distribution $\mathbb{D}$. For a given test input $x$, the attacker wins the game if the perturbation of the training data is small (as measured by an oracle $O(\mathbf{X}, \mathbf{X}_{\mathrm{adv}}) \mapsto \{0, 1\}$), and if the model misclassifies $x$.

The poisoning game in Figure 8a fails to capture an important facet of the facial recognition problem. The problem is not *static*: users continuously upload new pictures, and the model trainer actively scrapes them to update their model. Below, we introduce a *dynamic* version of the poisoning game, and show how a model trainer can use a *retroactive defense strategy* to win the game. In turn, we discuss how users and model trainers may *adapt* their strategies based on the other party's actions.

**Dynamic poisoning attacks.** To capture the dynamic nature of the facial recognition game, we define a generalized game for clean-label poisoning attacks in Figure 8b. The game now operates in rounds indexed by $i \geq 1$. In each round, the attacker perturbs new pictures and sends them to the defender. The strategies of the attacker and defender may change from one round to the next.

The game in Figure 8b allows for the data distribution $\mathbb{D}_i$ to change across rounds. Indeed, new users might begin uploading pictures, and users' faces may change over time. Yet, our thesis is that the main challenge faced by the user is precisely that *the distribution of pictures of their own face changes little over time*. For example, a facial recognition model trained on pictures of a user at 20 years old can reliably recognize pictures of the same user at 30 years old (Ling et al., 2010). Thus, in each round the defender can reuse training data $(\boldsymbol{\mathcal{X}}_{\mathrm{adv}}, \boldsymbol{\mathcal{Y}})$ collected in prior rounds. If the defender scrapes a user's images, the perturbations applied to these images cannot later be changed.

**Retroactive defenses.** The observation above places a high burden on the attacker. Suppose that in round $i$, the defender discovers a training technique $\texttt{train}_i$ that is resilient to *past* poisoning attacks $\texttt{Attack}_j$ for $j < i$. Then, the defender can train their model solely on data $(\boldsymbol{\mathcal{X}}_{\mathrm{adv}}, \boldsymbol{\mathcal{Y}})$ collected up to round $j$. From there on, the defender can trivially win the game by simply ignoring future training data (until they find a defense against newer attacks as well). Thus, the attacker's perturbations have to work against *all future defenses*, even those applied retroactively, for as long as the user's facial features do not naturally change. By design, this retroactive defense does not lead to an "arms race" with future attacks. The defender applies newly discovered defenses to *past* pictures only.

As we will show, this retroactive defense can even be instantiated by a fully *oblivious* model trainer, with no knowledge of users' attacks. The model trainer simply waits for a better facial recognition model to be developed, and then applies the model to pictures scraped before the new model was published. This oblivious strategy demonstrates the futility of preventing facial recognition with data poisoning, so long as progress in facial recognition models is expected to continue in the future.

**Adaptive defenses.** A model trainer that does not want to wait for progress in facial recognition can exploit another source of asymmetry over users: *adaptivity*. In our setting, it is easier for the

defender to adapt to the attacker, than vice-versa. Indeed, users must perturb their pictures *before* the model trainer scrapes them and feeds them to a secret training algorithm. As the trainer's model $f$ will likely be inaccessible to users, users will have no idea if their attack actually succeeded or not.

In contrast, the users' attack strategy is likely public (at least as a black-box) to support users with minimal technical background. For example, Fawkes offers open-source software to perturb images (Shan et al., 2021), and LowKey (Cherepanova et al., 2021b) and DoNotPay (Vincent, 2021) offer a Web API. The defender can thus assemble a dataset of perturbed images and use them to train a model. We call such a defender *adaptive*.[1]

**A note on evasion and obfuscation attacks.** The security games in Figure 8 assume that the training data is "clean label" (i.e., the user can still be identified in their pictures by other human users) and that the evaluation data is unperturbed. This is the setting considered by Fawkes (Shan et al., 2020) and LowKey (Cherepanova et al., 2021a), where a user shares their pictures online, but the user cannot control the pictures that are fed to the facial recognition model (e.g., pictures taken by a stalker, a security camera, or law enforcement).

The game dynamics change if the user evades the model with adversarial examples, by modifying their facial appearance at test time (Szegedy et al., 2013; Sharif et al., 2016; Thys et al., 2019; Gao et al., 2020; Cilloni et al., 2020; Rajabi et al., 2021; Oh et al., 2017; Deb et al., 2019; Browne et al., 2020; Deb et al., 2020). *Evasion* attacks favor the attacker: the defender must commit to a defense and the attacker can adapt their strategy accordingly (Tramer et al., 2020).

Our setting and security game also do not capture face *obfuscation* or *anonymization* techniques (Newton et al., 2005; Sun et al., 2018a;b; Sam et al., 2020; Cao et al., 2021; Maximov et al., 2020; Gafni et al., 2019). These attacks remove or synthetically replace a user's face, and thus fall outside of our threat model of clean-label poisoning attacks (i.e., the aim of these works is to remove identifying features from uploaded pictures, so that even a human user would fail to identify the user).

## 3 EXPERIMENTS

We evaluate two facial recognition poisoning tools, Fawkes (Shan et al., 2020) and LowKey (Cherepanova et al., 2021a), against various adaptive and oblivious defenses. We show that:

- An adaptive model trainer with black-box access to Fawkes and LowKey can train a robust model that resists poisoning attacks and correctly identifies all users with high accuracy.
- An adaptive model trainer can also *detect* perturbed pictures with near-perfect accuracy.
- Fawkes and LowKey are *already* broken by newer facial recognition models that appeared less than a year after the attacks were introduced.
- Achieving robustness against poisoning attacks need not come at a cost in clean accuracy (in contrast to existing defenses against adversarial examples (Tsipras et al., 2018)).

Code to reproduce our experiments is available at: `https://github.com/ftramer/FaceCure`.

### 3.1 ATTACKS

We evaluate three distinct poisoning attacks:

- *Fawkes v0.3*: this is the attack originally released by Fawkes (Shan et al., 2020) in July 2020. It received 500,000 downloads by April 2021 (Shan et al., 2021).
- *Fawkes v1.0*: this is a major update to the Fawkes attack from April 2021 (Shan et al., 2021).[2]
- *LowKey*: this attack was published in January 2021 with an acommpanying Web application (Cherepanova et al., 2021a;b).

---

[1]A black-box adaptive defense might be preventable with an attack that uses *secret per-user randomness* to ensure that robustness to an attack from one user does not generalize to other users. Existing attacks fail to do this, and designing such an attack is an open problem. Moreover, such an attack would remain vulnerable to our oblivious strategy.

[2]Unless noted otherwise, for our experiments with Fawkes, we use the most recent version 1.0 of the tool in its strongest protection mode ("high").

These three attacks rely on the same underlying principle. Each attack perturbs a user's online pictures with *adversarial examples* (Szegedy et al., 2013) so as to poison the training set of a facial recognition model. The attack's goal is that the facial recognition model learns to associate a user with spurious features that are not present in unperturbed pictures. Since the user does not know the specifics of the model trainer's facial recognition pipeline, the above attacks craft adversarial examples against a set of known facial recognition models (so-called *surrogate models*), in hopes that these adversarial perturbations will then *transfer* to other models (Papernot et al., 2016).

## 3.2 EXPERIMENTAL SETUP

The experiments in this section are performed with the *FaceScrub* dataset (Ng & Winkler, 2014), which contains over 50,000 images of 530 celebrities. Each user's pictures are aligned (to extract the face) and split into a training set (pictures that are posted online and scraped) and a test set, at a 70%-30% split. Additional details on the setup for each experiment can be found in Appendix B. We replicate our main results with a different dataset, PubFig (Kumar et al., 2009) in Appendix C.2.

**Attacker setup.** A user (one of FaceSrub's identities) perturbs all of their training data (i.e., their pictures uploaded online). We use Fawkes and Lowkey's official attack code in their strongest setting.

**Model trainer setup.** We consider a standard approach for facial recognition wherein the model trainer uses a fixed pre-trained feature extractor $g(x)$ to convert pictures into embeddings. Evaluation is done using a 1-Nearest Neighbor rule. Given a test image $x$, we find the training example $x'$ that minimizes $\|g(x) - g(x')\|_2$ and return the identity $y'$ associated with $x'$.

In Appendix C.1, we evaluate an alternative setup considered by Fawkes (Shan et al., 2020), where the model trainer converts the feature extractor $g(x)$ into a supervised classifier (by adding a linear layer on top of $g$) and fine-tunes the classifier on labeled pictures. This setting is less representative of large-scale facial recognition pipelines where the set of labels is not explicitly known.

**Feature extractors.** We consider a variety of pre-trained feature extractors that can be used by a model trainer to build a facial recognition system. We order the extractors chronologically by the date at which all training components necessary to replicate the model (model architecture, training data and loss function) were published.

- *FaceNet*: An Inception ResNet model pre-trained on VGG-Face2 (Schroff et al., 2015).
- *WebFace*: An Inception ResNet model pre-trained on CASIA-WebFace (Yi et al., 2014). This feature extractor is used as a surrogate model in the Fawkes v0.3 attack.
- *VGG-Face*: A VGG-16 model pre-trained on VGG-Face2 (Cao et al., 2018).
- *Celeb1M*: A ResNet trained on MS-Celeb-1M (Guo et al., 2016) with the ArcFace loss (Deng et al., 2018). This feature extractor is used as a surrogate model in the Fawkes v1.0 attack.
- *ArcFace*: A ResNet trained on CASIA-Webface with the ArcFace loss (Deng et al., 2018).
- *MagFace:* A ResNet trained on MS-Celeb-1M with the MagFace loss (Meng et al., 2021).
- *CLIP:* OpenAI's CLIP (Radford et al., 2021) vision transformer model, which we fine-tuned on CASIA-WebFace and VGG-Face2. Details on this model are in Appendix B.3. We are not yet aware of a facial recognition system built upon CLIP. Yet, due the the model's strong performance in transfer-learning and out-of-distribution generalization, it is a good candidate to test how existing attacks fare against facial recognition systems based on novel techniques (e.g., vision transformers and contrastive learning).

**Evaluation metric.** We evaluate the effectiveness of Fawkes and LowKey by the (top-1) error rate (a.k.a. protection rate) of the facial recognition classifier when evaluated on the *unperturbed* test images of the chosen user. We repeat each of our experiments 20 times with a different user in the position of the attacker, and report the average error rate across all 20 users.

## 3.3 ADAPTIVE DEFENSES

In this section, we assume that users perturb pictures using a public service (e.g., a Web application), to which the model trainer has *black-box* access. This assumption is realistic for Fawkes and LowKey, since both attacks offer a publicly accessible application. We show how the model trainer can *adaptively* train a feature extractor that resists these attacks.

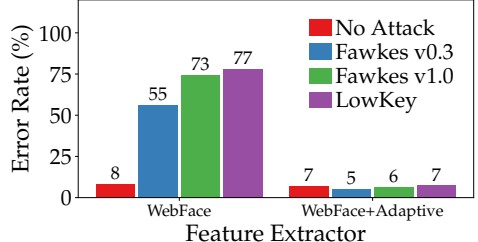

Figure 2: **Adaptive defenses break facial poisoning attacks.** Existing attacks break a standard WebFace model, but fail against a model explicitly trained on these attacks' perturbations.

**Training a robust feature extractor.** The model trainer begins by collecting a public dataset of unperturbed labeled faces $\mathbf{X}^{\text{public}}, \mathbf{Y}^{\text{public}} \leftarrow \mathbb{D}$ (e.g., a canonical dataset of celebrity faces), and calls the attack (as a black box) to obtain perturbed samples: $\mathbf{X}^{\text{public}}_{\text{adv}} \leftarrow \texttt{Attack}(\mathbf{X}^{\text{public}})$.

As the model trainer has access to both unperturbed images and their corresponding perturbed versions for a set of users, they can teach a model to produce similar embeddings for unperturbed and perturbed pictures of the same user—thereby encouraging the model to learn robust features. The hope then is that this robustness will generalize to the perturbations applied to other users' pictures.

We use the images of half of the FaceScrub users as the public labeled data ($\mathbf{X}^{\text{public}}, \mathbf{Y}^{\text{public}}$), and use the Fawkes and LowKey attacks as a black-box to obtain perturbed samples $\mathbf{X}^{\text{public}}_{\text{adv}}$. We robustly fine-tune the pre-trained WebFace feature extractor by adding a linear classifier head and then fine-tuning the entire model to minimize the cross-entropy loss on ($\mathbf{X}^{\text{public}}, \mathbf{Y}^{\text{public}}$) and ($\mathbf{X}^{\text{public}}_{\text{adv}}, \mathbf{Y}^{\text{public}}$). After fine-tuning, the classifier head is discarded.

The model trainer's adaptive strategy entails performing "robust data augmentation" for some users in the training set. We remark that this could also happen without explicit intervention from the model trainer. Indeed, some users are likely to have both perturbed and unperturbed pictures of themselves on the Web. (e.g., because they forgot to perturb some pictures, or because another user uploaded them). Feature extractors trained on these pictures would then be encouraged to learn robust features.

This robust training approach differs from *adversarial training* (Szegedy et al., 2013; Madry et al., 2017). Adversarial training makes a model robust against an attack *that depends on the model*. In our case, the attack is *fixed* (since the user has to commit to it), so the model trainer's goal is much easier.

**Results.** As a baseline, we first evaluate all attacks against a non-robust WebFace model. Figure 2 shows that the attacks are effective in this setting. For users who poisoned their online pictures, the model's error rate is 55-77% (as compared to only 8% error for unprotected users).

We now evaluate the performance of our robustly fine-tuned feature extractor. We use the extractor to obtain embeddings for the entire FaceScrub dataset, including the attacker's perturbed pictures (note that by keeping the users from $\mathbf{X}^{\text{public}}$ in the evaluation set, we favor the attacker as this only make the model's task *harder*). As shown in Figure 2, all attacks are rendered ineffective. The robust model's error rate for users who attempted to attack the system is no higher than for the average user.

In Appendix C.2, we show that our feature extractor retains its robustness for an alternative facial recognition benchmark. Thus, our robust training did not somehow overfit to the FaceScrub dataset.

**Attack detection.** With the knowledge of users' attack tool, the model trainer can also build a model to *detect* whether a given image has been perturbed. An accurate detector can be used to filter out perturbed images, and retain only unperturbed images of a user (if such pictures exist online). Moreover, detecting an attack attempt could itself be a privacy concern (e.g., law enforcement might actively target users whose attack attempts are detected).

For each attack, we fine-tune a standard pre-trained ImageNet model to distinguish between the perturbed and clean (unperturbed) images of 25 random users. We then evaluate the model on the perturbed and clean images of 20 other users not seen during training. As we show in Appendix B.4, perturbed pictures can be detected with near-perfect precision (99.8%) and recall (99.8%) for both the Fawkes and LowKey attacks. Note that the cost of a false positive is low, as this simply causes the model trainer to discard some user's unperturbed image.

### 3.4 OBLIVIOUS DEFENSES: TIME IS ALL YOU NEED

We now evaluate an even simpler "oblivious" defense strategy, wherein the model trainer *waits* for better facial recognition systems to be developed, and then retroactively applies such a system to pictures scraped in the past. To bypass this oblivious defense strategy, a poisoning attack must fool not only today's models, but also *all future models*. Asymmetrically, these new models do not have to be robust to all attacks; they just have to resist the specific attack that was used on prior pictures.

**Adversarial examples do not transfer across time.** Recall that attacks such as Fakwes and LowKey aim to build adversarial examples that transfer to the facial recognition model. The question of whether adversarial examples transfer *to future models* has received little attention. *We demonstrate that they do not*. We briefly depart from facial recognition to consider a standard vision task: ImageNet. The availability of a large number of pre-trained ImageNet models allows us to easily test how well adversarial examples created a few years ago transfer to today's models. Suppose that in 2015, a user took an ensemble of state-of-the-art models (at the time)—GoogLeNet, VGG-16, Inception-v3 and ResNet50—and generated adversarial examples against this ensemble using PGD (Madry et al., 2017). This setup mimics the attack used by LowKey (Cherepanova

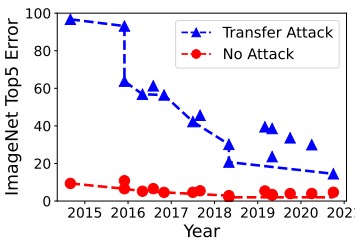

Figure 3: **Transferability of adversarial examples over time.** Each point is a model evaluated on the clean test set (red) and perturbed set (blue).

et al., 2021a). Figure 3 shows that the attack transfers well to contemporary models, but becomes near-ineffective for later models. Details on this experiment are in Appendix B.5.

Coming back to data poisoning, our thesis is that *attacks designed to transfer to today's models will inevitably decline as better models are developed*. Below, we provide evidence for this thesis.

**Results.** Figure 4 shows the performance of the Fawkes attack against a variety of feature extractors (each model was trained without knowledge of the attack). We order the models chronologically by the time at which all components required to train each model were publicly available.

We first observe that the original Fawkes v0.3 attack is completely ineffective. The only model for which it substantially increases the error rate is the WebFace model, which is the surrogate model that the attack explicitly targets. Thus, this attack simply fails at transferring to other feature extractors.

Fawkes' attack was updated in version 1.0 (Shan et al., 2021) to target the more recent Celeb1M feature extractor. The new version of Fawkes works perfectly against this specific feature extractor (error rate of 100%), and transfers to other canonical feature extractors such as VGG-Face, FaceNet and ArcFace. However, the Fawkes v1.0 attack fails against more recent extractors, such as MagFace and our fine-tuned CLIP model—thereby giving credence to our thesis.

Interestingly, we find that even if a model trainer uses a model that is vulnerable to Fawkes' new v1.0 attack, the 500,000 users who downloaded the original v0.3 attack cannot "regain" their privacy by switching to the updated attack. To illustrate, we show that if half a user's online pictures were originally poisoned with Fawkes v0.3, and half are later poisoned with Fawkes v1.0, the attack fails to break the Celeb1M model (6% error rate). Thus, once the model trainer adopts a model that resists past attacks, the protection for pictures perturbed in the past is lost—regardless of future attacks.

Figure 5 evaluates LowKey against the same set of feature extractors. LowKey fairs better than Fawkes and transfers to all canonical facial recognition models including MagFace. However, LowKey fails to break our fine-tuned CLIP model. While CLIP was not trained for facial recognition, it can extract rich facial features (Goh et al., 2021) and is remarkably robust to image perturbations (Radford et al., 2021). By fine-tuning CLIP on facial data[3], we achieve clean accuracy comparable to facial recognition models such as WebFace or VGG-Face, but with much higher robustness to attacks.

This experiment shows how developments in computer vision can break poisoning attacks that were developed before those techniques were discovered. Critically, note that CLIP is still vulnerable to adversarial examples. Thus, one could develop a new poisoning attack that explicitly targets CLIP,

---

[3]Following (Wortsman et al., 2021), we improved our fine-tuned model's robustness by linearly interpolating the weights of the fine-tuned model with the weights of the original CLIP model. Details are in Appendix B.5.

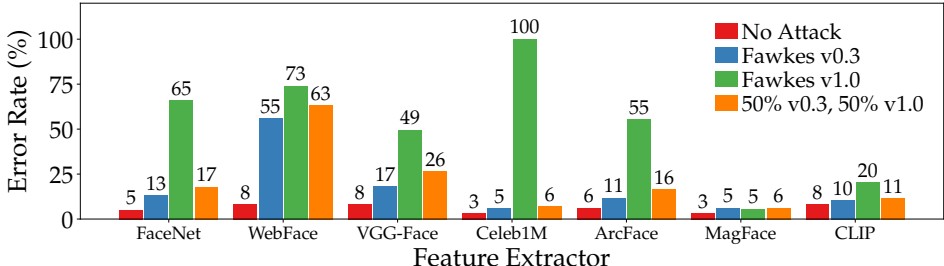

Figure 4: **Oblivious defenses break Fawkes**. Fawkes v0.3 does not transfer to any facial recognition model that it does not explicitly target. Fawkes v1.0 fares better, but fails against new models such as MagFace or CLIP. Moreover, a user that perturbs half their pictures with the original weak v0.3 attack and then switches to the stronger v1.0 attack cannot reclaim their privacy.

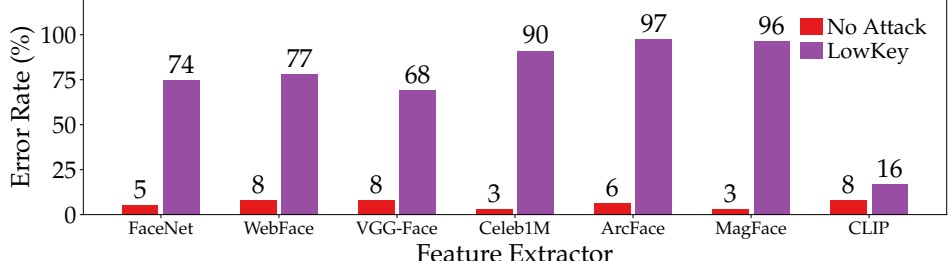

Figure 5: **Oblivious defenses can break LowKey**. The attack transfers well to canonical facial recognition models, but fails to transfer to our fine-tuned CLIP model.

and transfers to all models we consider. Yet, this attack will in turn fail when yet newer models and techniques are discovered. There is thus no arms-race here: the user always has to commit to an attack, and the model trainer later gets to apply newer and better models retroactively.

### 3.5 Increasing Robustness Without Degrading Utility

We have shown how a model trainer can adaptively train a robust feature extractor to resist known attacks, or wait for new facial recognition models that are robust to past attacks.

A potential caveat of these approaches is that increased robustness may come at a cost in accuracy (Tsipras et al., 2018). For example, our CLIP model is much more robust than other facial recognition models, but its clean accuracy is slightly below the best models. A model trainer might thus be reluctant to deploy a more robust model if only a small minority of users are trying to attack the system. We now show how a model trainer can combine a highly accurate model with a highly robust model to obtain the best of both worlds. We consider two potential approaches:

- *Top2*: If a facial recognition system's results are processed by a human (e.g., the police searching for a match on a suspect), then the system could simply run both models and return two candidate labels. The system could also run the robust model only when the more accurate model fails to find a match (which the human operator can check by visual inspection).

- Confidence thresholding: To automate the above process, the system can first run the most accurate model, and check the model's *confidence* (the embedding similarity between the target picture and its nearest match). If the confidence is below a threshold, the system runs the robust model instead. We set the threshold so that $< 2\%$ of clean images are run through both models.[4]

In Figure 6 we evaluate these two approaches for a facial recognition system that combines MagFace and a more robust model (either our adaptive feature extractor, or our fine-tuned CLIP). In both cases, the system's clean accuracy matches or exceeds that of MagFace, while retaining high robustness. Note that the MagFace model alone achieves $96.6\%$ top-1 accuracy and $96.8\%$ top-2 accuracy. Thus, remarkably, the top-2 accuracy obtained by outputting MagFace's top prediction and the robust model's top prediction is much better than outputting MagFace's top two predictions. This shows that the more robust models make very different mistakes than a standard facial recognition model.

---

[4]Confidence thresholding does lead to a short arms-race, that the attacker nevertheless loses (see Appendix D).

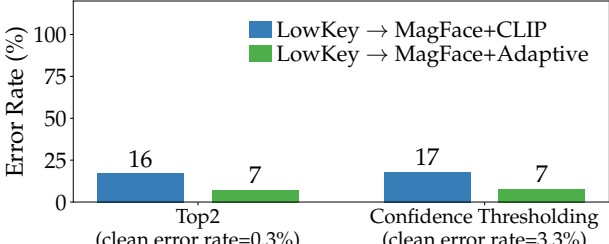

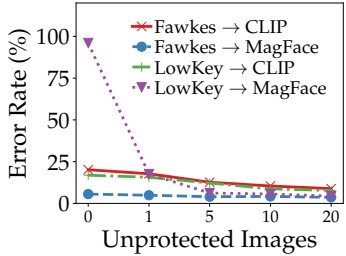

Figure 6: **A facial recognition system can use two models to achieve state-of-the-art accuracy and robustness**. (Left) A system that runs MagFace and a robust model *in parallel* has high top-2 accuracy and robustness; (Right) A system that runs the robust model if MagFace makes a non-confident prediction has high top-1 accuracy and robustness.

Figure 7: **Poisoning attacks fail when a few images are unprotected.** Uploading a single unprotected image online can significantly reduce a user's protection rate.

## 4 DISCUSSION

**Formal security games.** The motivation for systems such as Fawkes and LowKey is that adversarial examples can be used for poisoning and that these attacks transfer to different models. Yet, the unwritten assumption that these attacks also transfer to future (or adaptive) models does not hold. As beautifully argued by Gilmer et al. (2018), adversarial machine learning research often ignores important questions about the order in which parties interact, whether they can adapt, and how the game dynamics evolve over time. Defining formal security games, as we did in Section 2.2, is a useful way to reason about these questions, and we encourage future work to adopt this practice.

**Limitations.** Our evaluation uses curated datasets and a single attacking user. This may facilitate the model trainer's task compared to real applications. Yet, we also favor the attacker by measuring a model's top-1 accuracy, while a top-k match may work in some settings (e.g., a list of 50 candidates may suffice for the police to identify a suspect). Moreover, since all images in our experiments are pre-aligned and of fixed size, the attack does not have to transfer to an unknown pre-processing pipeline. Our main takeaway—that the defender has the upper hand—is oblivious to these experimental details.

**The game is already lost.** Taking a step back, we argue that even a "perfect" poisoning attack that works for all future models cannot save users' privacy. Indeed, many users already have unperturbed pictures online, which can be matched against new pictures for many years into the future.

Our experiments are conducted in the attacker-favorable setting where *all* of a user's training pictures are perturbed. The presence of unperturbed training pictures significantly weakens the efficacy of poisoning attacks (Shan et al., 2020; Evtimov et al., 2020). As we show in Figure 7, a user that uploads a single unperturbed picture (with all other pictures perturbed) already breaks current attacks.

Prior work recognized this issue and proposed collaborative poisoning attacks as a countermeasure (Shan et al., 2020; Evtimov et al., 2020). However, such attacks are futile for users whose unperturbed pictures are already online. There is a simple retroactive defense strategy for the model trainer: collect only pictures that were posted to the Web *before the first face-poisoning attacks were released* and match future pictures against these. Thus, for most users, the point in time where even a perfect poisoning attack would have stood a chance of saving their privacy is long gone.

## 5 CONCLUSION

Our work has demonstrated that poisoning attacks will *not* save users from large-scale facial recognition models trained on Web-scraped pictures. The initial motivation for these attacks is based on the premise that poisoning attacks can give rise to an "arms race", where better attacks can counteract improved defenses. We have shown that no such arms race can exist, as the model trainer can retroactively apply new models (obtained obliviously or adaptively) to pictures produced by past attacks. To at least counteract an oblivious model trainer, users would have to presume that no significant change will be made to facial recognition models in the coming years. Given the current pace of progress in the field, this assumption is unlikely to hold. Thus, we argue that legislative rather than technological solutions are needed to counteract privacy-invasive facial recognition systems.

**Ethics Statement.** Our work uncovers critical limitations of popular systems (Hill, 2020b; Marks, 2020; Heaven, 2021) that users turn to in the hope of protecting their privacy. In the spirit of responsible disclosure, we have informed the authors of Fawkes (Shan et al., 2020) and LowKey (Cherepanova et al., 2021a) of our results. We also mitigate the harm caused by our work by not targeting any real users of these systems.

As our experiments with oblivious defenses show, model trainers will obtain the upper hand regardless of the existence of our paper, merely by adopting more advanced facial recognition models. We thus view our work as a net *positive* to users' privacy, as it highlights the false sense of security that existing attacks provide.

Ultimately, we believe that the only viable course of action for privacy-conscious users is to avoid posting pictures online, or to support policy and legislation that restrict the use of facial recognition (Singer, 2018; Weise & Singer, 2020; Winder, 2020).

**Reproducibility Statement.** Our experiments rely on publicly available datasets and models, which are clearly referenced in the main text and described in more detail in Appendix B. We will release source code to reproduce all our experiments.

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

## A  SECURITY GAMES

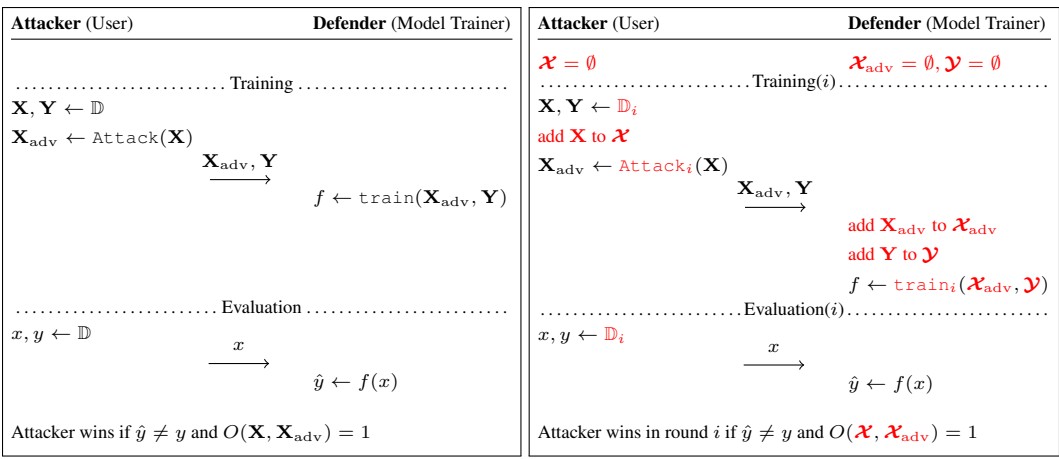

(a) **Game 1: Static game.** The attacker creates a clean-labeled poisoned training set $(\mathbf{X}_{\mathrm{adv}}, \mathbf{Y})$ and the defender trains a model $f$, which is evaluated on *unperturbed* inputs $x$. The attacker wins if $f$ misclassifies $x$ and the poisoned data $\mathbf{X}_{\mathrm{adv}}$ is "close" to the original data $\mathbf{X}$ (according to an oracle $O$).

(b) **Game 2: Dynamic game.** In each round $i \geq 1$, the attacker sends new poisoned data to the defender. The defender may train on *all* the training data $(\boldsymbol{\mathcal{X}}_{\mathrm{adv}}, \boldsymbol{\mathcal{Y}})$ it collected over prior rounds. The strategies of the attacker and defender may change between rounds.

Figure 8: **Security games for training-only clean-label poisoning attacks.**

## B  EXPERIMENTAL DETAILS

### B.1  GENERATION OF PERTURBED PICTURES.

For the experiments in Section 3, we generate perturbed images for random FaceScrub (Ng & Winkler, 2014) users using Fawkes (either version 0.3 in "high" mode[5] or the most recent version 1.0 in "high" mode[6]) and LowKey.[7]

We use the official pre-aligned and extracted faces from the FaceScrub dataset, and thus disable the automatic face-detection routines in both Fawkes and LowKey. For LowKey, we additionally resize all images to $112 \times 112$ pixels as we found the attack to perform best in this regime.

### B.2  ATTACK AND MODEL TRAINING SETUP

In each of our experiments, we randomly choose one user from the 530 FaceScrub identities to be the attacker. We perturb 100% of the training pictures of that user (70% of *all* pictures) with the chosen attack (Fawkes v0.3, Fawkes v1.0, or LowKey). The training set for the model trainer contains these perturbed pictures, as well as the training pictures of all other 529 FaceScrub users.

For evaluation, we extract features using a fixed pre-trained feature extractor and assign test points to the same class as the closest training point in feature space (under the Euclidean distance).

The FaceNet, VGG-Face and ArcFace models are taken from the DeepFace library.[8] The WebFace and Celeb1M models are taken from the released Fawkes tool (Shan et al., 2021). The MagFace model is taken from the official repository.[9] The CLIP model is taken from OpenAI's official repository[10] and fine-tuned using the procedure described in Appendix B.3.

---

[5]https://github.com/Shawn-Shan/fawkes/tree/63ba2f
[6]https://github.com/Shawn-Shan/fawkes/tree/5d1c2a
[7]https://openreview.net/forum?id=hJmtwocEqzc
[8]https://github.com/serengil/deepface
[9]https://github.com/IrvingMeng/MagFace
[10]https://github.com/openai/CLIP

To report the protection rate conferred by an attack (a.k.a. the model's error rate), we compute the model's error rate on the chosen user's unprotected test pictures. We then average these error rates across 20 experiments, each with a different random attacking user.

### B.3 FINE-TUNING CLIP FOR FACIAL RECOGNITION

OpenAI's pre-trained CLIP model achieves moderate accuracy as a facial recognition feature extractor. With the pre-trained ViT-32 model, a nearest neighbor classifier on extracted face embeddings achieves 83% clean accuracy using the FaceScrub dataset. While this accuracy is far below that of state-of-the-art models such as MagFace, CLIP's unique robustness to various image perturbations (Radford et al., 2021) acts as a strong defense against data poisoning: the model's error rate under the LowKey attack is only 29%.

To improve CLIP's performance for facial recognition, we fine-tune the model on two canonical facial recognition datasets, CASIA-WebFace (Yi et al., 2014) and VGG-Face2 (Schroff et al., 2015). For simplicity, we fine-tune CLIP using the same contrastive image-text loss as used for standard CLIP training. That is, we associate each face image with a text label "A photograph of user #*X*.", where *X* is a unique integer corresponding to each user in the dataset. (Alternatively, we could consider fine-tuning CLIP using a loss function tailored to facial recognition such as ArcFace or MagFace. Here, we were interested in evaluating a training procedure that is sufficiently *different* from existing facial recognition pipelines while still achieving strong accuracy.)

We fine-tune CLIP's pre-trained ViT-32 model on CASIA-WebFace and VGG-Face2 for 50 epochs using an open source implementation of CLIP training (Ilharco et al., 2021). The resulting model achieves high accuracy on FaceScrub (>95%), but loses most of CLIP's robustness against poisoning attacks. Similar behavior has been observed when fine-tuning CLIP for other tasks (Wortsman et al., 2021). Remarkably, Wortsman et al. (2021) recently showed that by *interpolating* the weights of the original CLIP model and the fine-tuned model, it is often possible to achieve high accuracy on the fine-tuned task while preserving CLIP's strong robustness properties. Concretely, given the pre-trained CLIP model with weights $\theta_{\text{CLIP}}$, and the fine-tuned model with weights $\theta_{\text{tuned}}$, we build a model with weights

$$\theta := \alpha \cdot \theta_{\text{tuned}} + (1 - \alpha) \cdot \theta_{\text{CLIP}} \, ,$$

where $\alpha \in [0, 1]$ is an interpolation parameter. We find that with $\alpha = 0.6$, we obtain a model that achieves both high accuracy on FaceScrub (92%) and high robustness against poisoning attacks (error rate of 16% against LowKey—half the initial error rate of the baseline CLIP model).

### B.4 ADAPTIVE DEFENSES

**Data generation using public attacks.** To train a robust feature extractor, we first generate perturbed pictures for many FaceScrub users using different attacks:[11]

Table 1: **Number of FaceScrub users whose images are perturbed for each attack.** Both the perturbed and unperturbed images of these users are used during robust training.

| Attack | Number of users |
|---|---|
| Fawkes v0.3 | 265 |
| Fawkes v1.0 | 50 |
| LowKey | 150 |

Note that this corresponds to 265 users in total (i.e., the users for the Fawkes v1.0 and LowKey attacks are a subset of the users for the Fawkes v0.3 attack). The public dataset $\mathbf{X}^{\text{public}}$ consists of the original pictures of these 265 users, and the perturbed dataset $\mathbf{X}^{\text{public}}_{\text{adv}}$ consists of all the perturbed pictures (across all attacks) of these users.

---

[11]We started this project by experimenting with Fawkes v0.3 and thus have generated many more perturbed pictures for that attack than for the newer attacks.

**Robust model training setup.** For the model trainer, we use the WebFace feature extractor from Fawkes that the original authors adversarially trained on a dataset different than FaceScrub (Shan et al., 2020). We first fine-tune this feature extractor on the data from the 265 chosen public users. That is, we add a 265-class linear layer on top of the feature extractor, and fine-tune the entire model end-to-end for 500 steps with batch size 32. To evaluate this robust feature extractor, we pick an attacking user at random (not one of the 265 public users), and build a training set consisting of the perturbed pictures of that user, and the unperturbed pictures of all other 529 users. We then extract features from this training set using the robust model and perform nearest neighbors classification.

**Attack detection.** To evaluate the detectability of perturbed pictures, we choose 45 users and generate perturbations using Fawkes v1.0 (in "low", "mid" and "high" protection modes) and LowKey. We use 25 users during training and 20 users during evaluation. For LowKey, we build a training dataset containing all unperturbed and perturbed pictures of the 25 users. For Fawkes, we do the same but split a user's perturbed pictures equally among its three attack strengths ("low", "mid" and "high"). Higher attack strengths introduce larger perturbations that provide more protection.

We then fine-tune a pre-trained MobileNetv2 model (Sandler et al., 2018) on the binary classification task of predicting whether a picture is perturbed. We fine-tune the model for 3 epochs using Adam with learning rate $\eta = 5 \cdot 10^{-5}$. The model is evaluated by its accuracy on the unperturbed and perturbed pictures of the 20 test users (each user has an equal number of perturbed and unperturbed pictures since we evaluate the Fawkes modes separately).

In Table 2 below, we report the detection accuracy, as well as precision, recall and AUC scores.

Table 2: **Performance of a model trained to detect perturbed images.** Detection performance is very high across all attacks even when smaller perturbations are used (i.e. Fawkes "low" and "mid").

| Attack | Detection Accuracy | Precision | Recall | AUC |
|---|---|---|---|---|
| Fawkes *high* | 99.8% | 99.8% | 99.8% | 99.99% |
| Fawkes *mid* | 99.6% | 99.8% | 99.4% | 99.91% |
| Fawkes *low* | 99.1% | 99.8% | 98.4% | 99.72% |
| LowKey | 99.8% | 99.8% | 99.8% | 99.97% |

Finally, we show that a detector that was trained on one system (i.e., Fawkes or LowKey) transfers to the other. That is, we take the detector model that was trained on 20 users perturbed with one attack, and evaluate whether this detector also succeeds in detecting the perturbations from the other attack.

Table 3: **Performance of a model trained to detect perturbed images of one attack (source) when evaluated on another attack (destination).**

| Source → Destination | Detection Accuracy | Precision | Recall | AUC |
|---|---|---|---|---|
| Fawkes → LowKey | 99.4% | 99.0% | 99.8% | 99.59% |
| LowKey → Fawkes *high* | 71.9% | 100% | 43.9% | 98.3% |

## B.5 OBLIVIOUS DEFENSES

To generate Figure 4, we perturb one user's training pictures using either the Fawkes v0.3 attack, the Fawkes v1.0 attack, or a joint attack that perturbs half the user's training pictures with either of the two Fawkes versions. We then use each pre-trained feature extractor to extract embeddings for the entire training set of 530 users, and evaluate the performance of a 1-nearest neighbor classifier on the user's unprotected test images. To generate Figure 5, we repeat the same process with the LowKey attack.

**Transferability of adversarial examples across time.** For the experiment in Figure 3 in Section 3.4, we evaluate the transferability of adversarial examples crafted using an ensemble of ImageNet models from 2015 to future ImageNet models.

We create adversarial examples for an ensemble of GoogLeNet, VGG-16, Inception-v3 and ResNet-50 models using the PGD attack of Madry et al. (2017) with a perturbation budget of $\epsilon = 16/255$. The attack lowers the ensemble's top-5 accuracy from 93% to 0%.

We then transfer these adversarial examples to a variety of more recent ImageNet models. All pre-trained models are taken from either `pytorch/vision`[12] or `wightman/pytorch-image-models`.[13] For each model, we report the year the model was originally proposed, and the top-1/top-5 accuracy on ImageNet and on the transferred adversarial examples in Table 4

Table 4: **Transferability of adversarial examples from an ensemble of four models— GoogLeNet, VGG-16, Inception-v3, ResNet-50—to future ImageNet models.** Numbers in bold show the models that are most robust to the attack at a given point in time.

| Model | Year | Top-1 Acc | | Top-5 Acc | |
|---|---|---|---|---|---|
| | | Clean | Adv | Clean | Adv |
| ResNet-101 (He et al., 2016) | 2015-12 | 76% | **8%** | 93% | **36%** |
| Wide-ResNet-101 (Zagoruyko & Komodakis, 2016) | 2016-05 | 78% | **14%** | 95% | **43%** |
| Densenet-121 (Huang et al., 2017) | 2016-08 | 77% | 11% | 94% | 39% |
| ResNeXt-101_32x8d (Xie et al., 2017) | 2016-11 | 79% | **15%** | 95% | **44%** |
| Dual Path Networks-107 (Chen et al., 2017) | 2017-07 | 80% | **25%** | 95% | **58%** |
| SE-ResNeXt101_32x4d (Hu et al., 2018) | 2017-09 | 80% | 23% | 95% | 54% |
| IG-ResNeXt-101_32x16 (Mahajan et al., 2018) | 2018-05 | 85% | **38%** | 97% | **70%** |
| WSL-ResNext-101_32x48d (Mahajan et al., 2018) | 2018-05 | 85% | **49%** | 98% | **79%** |
| SK-ResNeXt-50_32x4d (Li et al., 2019) | 2019-03 | 79% | 29% | 94% | 61% |
| SSL-ResNeXt-101_32x16d (Yalniz et al., 2019) | 2019-05 | 82% | 26% | 97% | 61% |
| SWSL-ResNeXt-101_32x16d (Yalniz et al., 2019) | 2019-05 | 82% | 44% | 96% | 76% |
| ECA-ResNet-101s (Wang et al., 2020) | 2019-10 | 81% | 34% | 96% | 66% |
| ResNeSt-101 (Zhang et al., 2020) | 2020-04 | 79% | 38% | 96% | 70% |
| ViT-16 (Dosovitskiy et al., 2020) | 2020-10 | 83% | **57%** | 97% | **86%** |

## C  ADDITIONAL EXPERIMENTS

### C.1  SUPERVISED FACIAL RECOGNITION CLASSIFIERS

In Section 3, we evaluated a standard facial recognition pipeline based on nearest neighbor search on top of facial embeddings. For completeness, we evaluate two additional facial recognition setups considered by Fawkes (Shan et al., 2020), where a face classifier is trained in a supervised fashion on a labeled dataset of users' photos.

Specifically, given a pre-trained feature extractor $g(x)$, we add a linear classifier head on top of $g$, and then train the classifier to minimize the cross-entropy loss on the full FaceScrub dataset (i.e., the training data of all 530 users). We consider two approaches:

- *Linear*: the weights of the feature extractor $g$ are frozen and only the linear classification head is tuned;
- *End to end*: the feature extractor and the linear classifier are jointly tuned on the training dataset.

**Baseline.** We first evaluate the baseline performance of the Fawkes (v1.0) and LowKey attacks for linear fine-tuning and end to end tuning. We also reproduce the results for nearest neighbor search for comparison. As shown in Figure 9a, the poisoning attacks are more effective when the model trainer fine-tunes a linear classifier on top of a fixed feature extractor (error rate $\geq 93\%$) instead of fine-tuning the entire classifier end-to-end, or performing nearest neighbor search in feature space

---

[12]https://github.com/pytorch/vision
[13]https://github.com/rwightman/pytorch-image-models

(error rates of 73–77%). That is, linear classifiers are unsurprisingly easier to poison given their lower capacity.

**Robust training.** We also replicate our experiments with an adaptive model from Section 3.3. When the model trainer uses linear fine-tuning, they use the robust feature extractor described in Section 3.3 and fine-tune a linear classifier on top. When the model trainer fine-tunes a model end-to-end, we add pairs of public unperturbed and perturbed faces $(\mathbf{X}_{\text{adv}}^{\text{public}}, \mathbf{Y}^{\text{public}})$ to the model's training set, and tune the entire classifier end to end.

As shown in Figure 9b, each of the three facial recognition approaches we consider can be made robust. In all cases, the user's protection rate (the test error rate on unperturbed pictures) is similar to the classifier's average error rate for unprotected users.

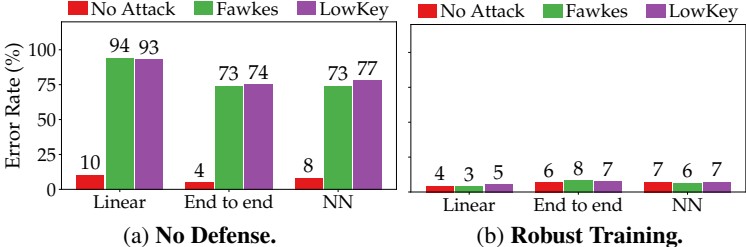

Figure 9: **Adaptive defenses against Fawkes and LowKey.** We report (a) the baseline performance (i.e. when no defense is used) for three training modes (Linear, End to end, Nearest neighbors); (b) the attack performance after robust training.

## C.2    Experiments on PubFig

In this section, we replicate the results from Section 3 on a different facial recognition dataset. We use a curated subset of the PubFig dataset (Kumar et al., 2009), with over 11,000 pictures of 150 celebrities.

Figure 10 and Figure 11 show the protection rates of the Fawkes v0.3, Fawkes v1.0 and LowKey attacks against the various feature extractors we consider. Note that the adaptive feature extractor used here is the same one as in Section 3.3, which was trained without any data from PubFig.

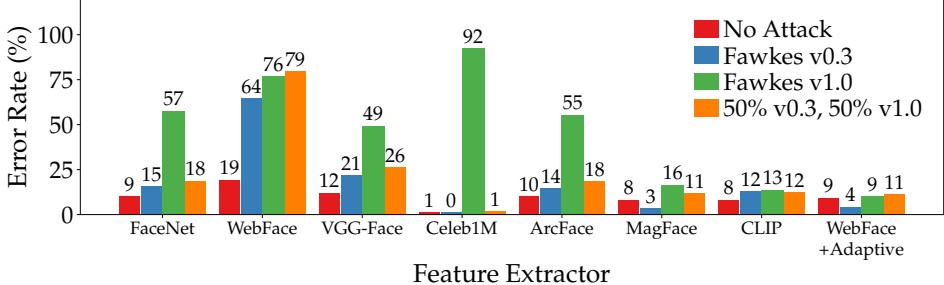

Figure 10: **Adaptive and oblivious defenses break Fawkes on PubFig.** As on FaceScrub in Figure 2 and Figure 4, the Fawkes v0.3 attack fails to transfer, and the Fawkes v1.0 attack fails against new models such as MagFace or CLIP as well as against an adaptively trained model.

The results on PubFig are qualitatively similar as on FaceScrub:

- The Fawkes v0.3 attack fails to transfer to models other than the WebFace model that the attack explicitly targets.
- The Fawkes v1.0 attack transfers reasonably well to older facial recognition models, but is ineffective against MagFace and CLIP.
- The LowKey attack works well against all "traditional" facial recognition models, but fails against our fine-tuned CLIP model.

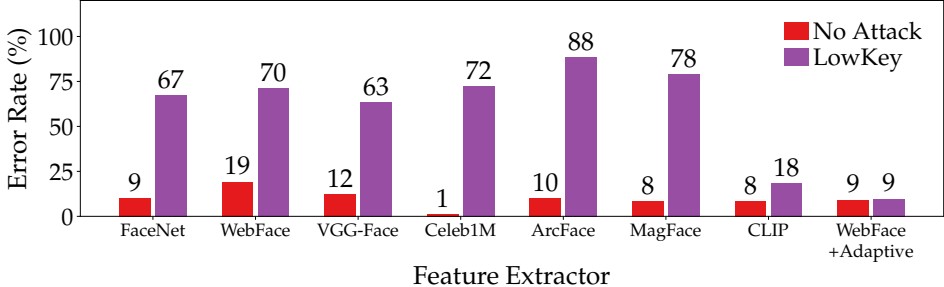

Figure 11: **Adaptive and oblivious defenses break LowKey on PubFig**. As on FaceScrub in Figure 2 and Figure 5, the LowKey attacks transfers well to canonical pre-trained facial feature extractors, but fails against our CLIP model and against an adaptively trained model.

- All attacks are ineffective against the robust feature extractor.

In Figure 12, we further replicate the experiment from Section 3.5 on building a facial recognition system that combines state-of-the-art accuracy and robustness. As on FaceScrub, a system that only returns confident predictions from the accurate model, and otherwise diverts to a more robust model, achieves strong accuracy and robustness. Note that a system that solely uses the MagFace model achieves a clean error rate of 8.1% (top-1) and 6.2% (top-2).

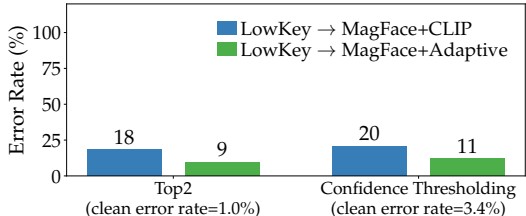

Figure 12: **Replication of Figure 6 on PubFig**. (Left) A system that runs both MagFace and a robust model (either CLIP or our adaptive model) in *parallel* achieves high top-2 accuracy and robustness; (Right) A system that only runs the robust model when MagFace fails to make a confident prediction has high top-1 accuracy and robustness.

## D   AN ARMS-RACE ON CONFIDENCE THRESHOLDING

In Section 3.5, we showed how to build a facial recognition system that combines a highly accurate model and a highly robust model. The system first runs the most accurate model. If the model's prediction has low confidence, the system instead runs the robust model.

The reason that confidence thresholding works is that Fawkes and LowKey are *untargeted* attacks. That is, each of the user's training pictures is perturbed to produce embeddings that are far from the clean embeddings. These perturbed embeddings will typically be far from *all* facial embeddings, and a model will thus not find a close match when evaluated on an unperturbed picture of the user.

A user could thus aim to circumvent the confidence thresholding system by switching to a *targeted* attack. This actually requires that users collude: user A would perturb their pictures so as to match the clean embeddings of user B, so that an unperturbed picture of user B would get mislabeled as user A with high confidence.

The model trainer can further counteract such an attack. The model trainer first runs the target image through the most accurate model. If the model returns a confident match, the system further checks whether the returned match is a *perturbed* image (by using an attack detector as described in Section 3.3 and Appendix B.4). If this is the case, the system ignores the accurate model's prediction and runs the target image through the more robust model instead.

To circumvent this adapted system, users would have to not-only collude to create targeted attacks, but also ensure that these attacks fool the model trainer's detector. But since users have to commit to an attack *before* the model trainer decides on a defense strategy, users will always be on the losing end of this cat-and-mouse game.

