# OpenReview forum: "Data Poisoning Won’t Save You From Facial Recognition"
_ICLR.cc/2022/Conference — ICLR 2022 Poster_

### Official Review · Reviewer_6s7m · 2021-10-28

**Correctness:** 3
**Technical Novelty And Significance:** 2
**Empirical Novelty And Significance:** 2
**Recommendation:** 1
**Confidence:** 5

**Main Review:**

+ The insight of this paper is very useful for the proctecter of user face.
- It is not clear how the oblivious trainer works.
- There is no clear presentation about the poisoning details of Flawkes and LowKeys.
- The technical novelty is very limited. Rather than fancying legislative alternative, a research paper needs to propose technical solution.


**Summary Of The Paper:**

This paper studies the effect of data poisoning in face recognition and the relation to the defense techniques. Conventionally, the poisoned data will fail the face recognition models who is trained without defense strategy. Two solutions of defense are given: oblivious trainer and adaptive trainer. The claim is that, any existing poisoning methods cannot protect the privacy of users in the face images.

**Summary Of The Review:**

Overall, this is a facial privacy analysis with insightful claims, but the presentation and the discussion is very confusing. Thus, we are not able to agree this argument whether is reasonable and solid. The final rating will depend on the authors’ feedback.

---

> ### Author Response · Authors · 2021-11-14
> **Response**
>
> We thank the reviewer for their feedback.
>
> **Unfortunately, the current review offers very limited actionable criticism of the paper. The reviewer claims the paper is “confusing”. What parts of the paper would the reviewer want us to clarify?**
>
> Below, we address the points raised by the reviewer.
>
> > *“It is not clear how the oblivious trainer works.”*
>
> The process is detailed in Section 2.2 and again in Section 3.4. Which part would the reviewer want us to clarify? The oblivious trainer simply collects facial data and trains an undefended model.
>
> > *“There is no clear presentation about the poisoning details of Flawkes and LowKeys”*
>
> The general principle behind the attacks is explained in Section 3.1. The exact details of what the attacks do is not important for our purpose, and we thus refer the interested reader to the original papers.
>
> But, we’re happy to provide some additional details on these attacks in our final version if the reviewer could clarify which aspects of these systems could merit further discussion.
>
> > *“The technical novelty is very limited. Rather than fancying legislative alternative, a research paper needs to propose technical solution.”*
>
> Could the reviewer point us to the section of the call-for-papers that mandates that a research paper must propose a technical solution to a problem that the paper identifies?
>
> ICLR’s call-for-papers reads: *“the scope includes "societal considerations of representation learning including fairness, safety, privacy, and interpretability".*
> **We believe that our paper falls well within this scope.**
>
> There are many papers that highlight issues in ML without being able to propose an actionable solution. For example, papers that show that ML models are: (1) vulnerable to adversarial examples; (2) not robust to various distribution shifts; (3) biased against various subgroups, etc.
>
> It is also not clear to us that a technical solution to the problem of large-scale facial recognition necessarily exists. Humans are very good at performing facial recognition and thus, in principle, it should be possible to automate this procedure with machine learning. In this sense, the attacks that we evaluate are essentially trying to defer the inevitable (and we show that they will fail at doing that even against oblivious defenders).
>
> This is why we argue that a technical solution may be hopeless and that legislation may be the only way to prevent abuses of facial recognition.
>
> Finally, we do consider the following as novel technical contributions in our paper:
> - The formal game analysis of dynamic poisoning attacks in Section 2, which clarifies some subtle differences between the attack dynamics of evasion and poisoning attacks.
> - A rigorous evaluation of various defense techniques against poisoning attacks.
> - The design of defense strategies that achieve high robustness and accuracy, in Section 3.5

---

> > ### Comment · Reviewer_6s7m · 2021-11-19
> > **Incomplete survey&experiments and technical absence**
> >
> > (1) As there is no clear definition of oblivious defense in this paper, we are fairly confused about the defense, and find it extremely simple as waiting for better face recognition models which is a primary version of purification. Besides, there are also many existing works to anonymize facial images which do not appear in this paper as the role of attacker/protector. So, in this part, the problem should be the missing important reference and important experiments.
> > Also, there is no formal definition of $X$ and $Y$.
> >
> > (2) We clarify that these systems should be presented with more details so that they can be compared with the existing facial identity anonymizing methods which can work well to fool the state-of-the-art face recognition models without significant visual modification.
> >
> > (3)
> >
> > Firstly, let us clarify that we do recognize the major tech contribution of this paper which is that the authors find that the existing poisoning methods fail to protect the facial identity. This finding, however, is based on the incomplete survey and experiment (see above). The poisoning/protectors are definitively hopeless or not, may the authors provide more solid verification about this?
> >
> > Secondly, the contribution of this paper, thus, narrows down to the second one, which is appealing the legislative roundabout. Considering this is a legislative-only claim as the remaining contribution, we suggest the authors to submit this paper to the conferences/journals related to the sciences of law.
> >
> > Finally, sure, banning human face recognition is the MAINSTREAM sense nowadays. Making contribution to the mainstream is understandable. Yet, the challenge is the way we contribute. Should we ban the using of current water for preventing inundation, or develop our technology and invent a hydropower station?

---

> > > ### Comment · Reviewer_6s7m · 2021-11-19
> > > **References**
> > >
> > > Facial Identity Anonymizing
> > >
> > > [1]Cao J, Liu B, Wen Y, et al. Personalized and Invertible Face De-Identification by Disentangled Identity Information Manipulation[C]//Proceedings of the IEEE/CVF International Conference on Computer Vision. 2021: 3334-3342.
> > >
> > > [2]Maximov M, Elezi I, Leal-Taixé L. Ciagan: Conditional identity anonymization generative adversarial networks[C]//Proceedings of the IEEE/CVF Conference on Computer Vision and Pattern Recognition. 2020: 5447-5456.
> > >
> > > [3]Gafni O, Wolf L, Taigman Y. Live face de-identification in video[C]//Proceedings of the IEEE/CVF International Conference on Computer Vision. 2019: 9378-9387.
> > >
> > >
> > > Purification
> > >
> > > [1]Deb D, Liu X, Jain A K. FaceGuard: A Self-Supervised Defense Against Adversarial Face Images[J]. arXiv preprint arXiv:2011.14218, 2020.
> > > [2]Agarwal A, Singh R, Vatsa M, et al. Image transformation based defense against adversarial perturbation on deep learning models[J]. IEEE Transactions on Dependable and Secure Computing, 2020.

---

> > > > ### Author Response · Authors · 2021-11-19
> > > > **Response**
> > > >
> > > > We thank the reviewer for clarifying their review of our paper. We answer the reviewer's three comments below, and have edited our submission accordingly (the changes are marked in blue).
> > > >
> > > > (1)
> > > > The oblivious defense *is* extremely simple. That's our whole point, that even just waiting for newer facial recognition systems will break current poisoning attacks.
> > > >
> > > > We thank the reviewer for linking to related work on face anonymization and purification. We have added a discussion in Section 2.2 ("A note on evasion and obfuscation attacks") where we explain that these works consider a very different threat model than ours:
> > > >
> > > > - anonymization approaches are *not* clean-label (and they are also not necessarily poisoning attacks). Their aim is to make faces non-identifiable (even for humans). Our defenses and our take-aways do not apply to such approaches.
> > > > - the two linked papers for purification are defenses against *evasion* attacks: here it is assumed that a user's training pictures are unaltered, but the test-time pictures are perturbed. This is again a very different threat model for which we make no claims.
> > > >
> > > > Clean-label poisoning attacks are very compelling because they still let users upload pictures online so as to share them with others, and they don't require users to have control over the pictures that are ultimately fed to the facial recognition system. This makes these attacks very user-friendly, and explains why attacks such as Fawkes have had a huge success in building a user base.
> > > >
> > > > Regarding the definition of $X, Y$, these are standard notations used for features and labels. We have added a note clarifying that in our setting, $X$ represents the facial images of users, and $Y$ the corresponding labels. The formal analysis in Section 2.2. is not limited to facial recognition though, and applies to any clean-label poisoning attack.
> > > >
> > > > (2)
> > > > We hope that the above distinction between poisoning, evasion and anonymization helps to clarify how Fawkes and LowKey differ from other approaches. The setting considered in our paper and in Fawkes & LowKey is that a user perturbs the pictures they post online (but minimally, so that their friends can still identify them), with the goal that machine learning models trained on these pictures will fail to identify unperturbed pictures of the user. This setup is summarized in Figure 1.
> > > >
> > > > Our paper shows that such poisoning attacks will not work. We make no claims about evasion attacks or face anonymization techniques, as these correspond to very different threat models.
> > > >
> > > > (3)
> > > > Our claim about the need for legislation is not a contribution of our paper, and we never say that it is. This is simply a concluding thought we make (once in the introduction and once in the conclusion), based on our technical evaluation.
> > > >
> > > > More specifically, we note that if users want to continue uploading identifiable pictures of themselves online, we don't have a technical way of preventing facial recognition systems from being developed (and thus legislation may be the only alternative).
> > > > Whether this position happens to be "mainstream" or not is irrelevant.

---

> > > > > ### Author Response · Authors · 2021-11-30
> > > > > **Further questions?**
> > > > >
> > > > > Could the reviewer please clarify the rationale for reducing their score to a strong reject following our rebuttal?
> > > > >
> > > > > We have clarified our paper's scope and claimed contributions.
> > > > > Does the reviewer disagree with these, or are there other points we could aim to address?

---

### Official Review · Reviewer_3JiV · 2021-10-29

**Correctness:** 3
**Technical Novelty And Significance:** 3
**Empirical Novelty And Significance:** 2
**Recommendation:** 8
**Confidence:** 3

**Details Of Ethics Concerns:**

I do not find any ethical concern about this work. It points out the flaws of the previous methods and informs users not to have a false sense of security by using those methods.

**Main Review:**

### Strengths
- This paper defines the dynamic game scenario, which is more practical than the static game scenario used in the previous poison-based identity protection works
- It proposes two defenses that can defeat the poison-based identity protection methods
- The authors propose a strategy for the model trainer to defense from poisoning attacks without sacrifying accuracy
- The authors give a good discussion on the flaws of the poison-based identity protection approach. The game is already lost, and users should not rely on the methods in that category to have a false sense of security.
- This paper has a good amount of references

### Weaknesses
- In the adaptive defense, the authors assume access to the poisoning function, which may be unrealistic. The poisoning function should be secret for a better protection,
- Obvious defense can break LowKey only once (Figure 6). There is no guarantee that robust models, like CLIP, will be popular in the future facial recognition models.

### Not a weakness of the paper itself
I agree with the section "The game is already lost": the poison-based identity protection approach is very unrealistic and has a low value.

**Summary Of The Paper:**

Recent works propose to protect users from facial recognition by poisoning their images before uploading them to the Internet (called poisoning "attacks"). This paper reveals the flaws of these methods by designing two effective methods (called "defenses") to defeat that protection mechanism. The first approach is adaptive defense, in which the model trainer assumes to have access to the poisoning function as a black-box. He then can collect a clean facial image dataset, create perturbed and unperturbed versions, and finetune the poisoned face recognition model to learn robust features. The second approach, called obvious defense, relies on the fact that poisoned examples do not transfer well over time to newer models. Hence, the model trainer can wait to obtain a better face recognition model and properly safely finetune it on the perturbed images. Both methods successfully defeat two poisoning attack baselines, raising awareness on the inefficiency of the poisoning-based identity protection mechanism.

**Summary Of The Review:**

This paper reveals the flaws of previous works that use poison attacks to protect user identity from face recognition systems. It has some good discussions and provides two methods to break those protections. There are still some concerns about the experiments and the significance of the previous works the paper is targeting. Hence, I give it a Borderline, slightly towards Acceptance.

---

> ### Author Response · Authors · 2021-11-14
> **Response**
>
> We thank the reviewer for their valuable comments and for their positive assessment of our paper. Below, we address the reviewer's main points.
>
> > *“In the adaptive defense, the authors assume access to the poisoning function, which may be unrealistic. The poisoning function should be secret for a better protection”*
>
> A secret poisoning function would indeed be harder to adapt to, yet this would also make it harder for systems such as Fawkes or LowKey to protect the privacy of a large number of users.
>
> An individual tech-savvy user may decide to develop a secret attack. In this case, our oblivious defenses would obviously still apply.
> We also find that our adaptive defenses transfer moderately between attacks. For example, in Appendix A.4 we show that it is possible to build a robust detector for the attacks of Fawkes or LowKey, without requiring access to the poisoning function.
>
> We will include a more detailed discussion of secret attacks in our final paper.
>
> > *“There is no guarantee that robust models, like CLIP, will be popular in the future facial recognition models”*
>
> Our point here is that future facial recognition models **could** use CLIP, in which case all users’ privacy would be lost. The existence of even a **single** defense against a poisoning attack invalidates any guarantees that the attack can provide (since it is the defender that gets to adapt to the attack, and not vice-versa).
>
> Moreover, users typically don't know what type of models are being used by facial recognition companies. So a user of current attacks would have to "blindly" hope that models such as CLIP won’t be used now or in the future.
>
> > “There are still some concerns about the experiments and the significance of the previous works the paper is targeting”
>
> Fawkes got a very significant amount of media attention and a large user base (500k downloads according to the authors). In this sense, Fawkes is orders-of-magnitude more significant than *any* other ML defense that has been evaluated in the past (e.g., there exists a large body of work on evaluating various defenses against adversarial examples, but to our knowledge none of these defenses have actually ever been deployed in practice).
>
> Moreover, we note that LowKey was published at ICLR 2021 and we thus argue that a paper that thoroughly evaluates this system should be of significant interest for ICLR 2022.

---

> > ### Comment · Reviewer_3JiV · 2021-11-18
> > **I have increased your score**
> >
> > I find your answer reasonable. Hence, I have increased your score to 8.

---

### Official Review · Reviewer_8c4m · 2021-11-02

**Correctness:** 4
**Technical Novelty And Significance:** 2
**Empirical Novelty And Significance:** 2
**Recommendation:** 6
**Confidence:** 4

**Main Review:**

### Strengths:
1. The paper is well written and easy to follow.
2. The topic discussed in this paper is of significant importance as the rapid development of deep learning techniques nowadays also poses great threats to people's privacy, especially for the face data.
3. Extensive experiments have been conducted to statistically ascertain the authors' claims.

### Weaknesses:
From technical point of view, there is less innovation in this paper. It is widely known that adversarial attacks, either obtained in white-box or black-box manners, can be effectively guarded by robust training or by adopting a different algorithm. The experimental results are within expectation and little technical insight is gained.


**Summary Of The Paper:**

This paper points out that current data poisoning techniques cannot effiectively protect users privacy, i.e., face data, on the Internet. The authors have examined several strategies to enable modern face recognition models to defense attacks from widely used data poisoning methods. Experimental results suggest that those data poisoning attacks can be easily defensed by adaptively tuning the face recognition models or using more advanced algorithms which would be developed in the future. The main conclusion is that people should not rely on technical solutions to protect users privacy and legislation actions are what is actually needed.

**Summary Of The Review:**

This paper has addressed an important problem that, data poisoning techniques are insufficient to protect people's privacy from modern development of face recognition models and it is of great importance to draw people's attention to the discussed topic, to potentially push any legislation actions instead of searching for technical solutions to protect users privacy. However, I think the technical contribution from this paper is limited as little new insight is provided.

---

> ### Author Response · Authors · 2021-11-14
> **Response**
>
> We thank the reviewer for their valuable comments. We argue that the contributions of our paper (both technical and conceptual) are significant and can contribute to a better understanding of the limits of adversarial ML.
>
> > *“It is widely known that adversarial attacks, either obtained in white-box or black-box manners, can be effectively guarded by robust training or by adopting a different algorithm. The experimental results are within expectation and little technical insight is gained.”*
>
> While our results might not be surprising in hindsight, systems such as Fawkes and LowKey were published at top venues in security and machine learning (USENIX Security 2020 and ICLR 2021), and have attracted a lot of media attention and real users. So presumably it is not obvious to everyone that these systems are ineffective. A study like ours is thus important to underline the limitations of poisoning attacks in order to avoid luring users into a false sense of security.
>
> We further consider the following as novel technical contributions in our paper:
> - The formal game analysis of dynamic poisoning attacks in Section 2, which clarifies some subtle differences between the attack dynamics of evasion and poisoning attacks.
> - A rigorous evaluation of various defense techniques against poisoning attacks. The original evaluations performed by the Fawkes and LowKey papers were deemed sufficient by reviewers at the time. We believe our paper thus makes an important contribution in demonstrating how such evaluations may have overestimated the protections offered to users.
> - The design of defense strategies that achieve high robustness and accuracy, in Section 3.5. These strategies demonstrate that robustness need not come at a cost in accuracy in all situations, as is often claimed in the adversarial examples literature.

---

> > ### Comment · Reviewer_8c4m · 2021-11-29
> > **A good point have been made in the rebuttal**
> >
> > The authors have made a good point that the limitation of adversarial attack may not be so obvious to non-professional users without relevant background and therefore existing attack systems like Fawkes and LowKey may have misled ordinary users to some extent. I agree that there should be some different voice on such an influential platform like ICLR. Therefore, I am willing to increase my rating by one level.
> >
> > However, I still think the claimed technical contributions are not very significant and the main value of this paper is pointing out the fragility of existing data poisoning techniques by conducting extensive experiments in order to raise more attention from the public.

---

### Official Review · Reviewer_6oYM · 2021-11-02

**Correctness:** 3
**Technical Novelty And Significance:** 3
**Empirical Novelty And Significance:** 3
**Recommendation:** 8
**Confidence:** 5

**Main Review:**

Strengths
This is an interesting paper. The evaluation criteria and the game setup was well conducted. A large number of feature extractors were evaluated showing the problem is consistent across multiple. It is especially interesting to see how even "oblivious" systems can just wait and see to get better. Figure 5 is shows how Fawkes deteriorates over time with new extractors causing it to fail. The paper is well-written and easy to read. Ethical concerns are well documented in ethics statement.

Weaknesses
This is a strong statement "Ultimately, we believe that the only viable course of action for privacy-conscious users is to avoid posting pictures online, or to support policy and legislation that restrict the use of facial recognition (Singer, 2018; Weise & Singer, 2020; Winder, 2020)." and may not be a feasible solution to the problem, although supporting policy is certainly feasible. A more concrete solution to the problem would have strengthened the paper.

Minor point: The paper argues that it is not an arms-race, however, it can still be considered an arms race with the constraint that the defender is in a better position to win. This is arguably just a semantic difference, and a minor point.

**Summary Of The Paper:**

Paper presents an analysis of two systems for poisoning attacks. The paper shows that perturbing facial images does not offer long term security; future systems can still recognize the once perturbed image(s).

**Summary Of The Review:**

The paper does a good job analyzing poisoning attacks. It shows that the methods fail over time and they do not generalize to future attacks. It is in interesting paper and has value for the community to consider longer term security measures for facial recognition privacy concerns.

---

> ### Author Response · Authors · 2021-11-14
> **Response**
>
> We thank the reviewer for their very positive assessment of our paper and their insightful comments. Below we address the main points raised by the reviewer.
>
> > *“A more concrete solution to the problem would have strengthened the paper.”*
>
> It is not clear to us that a technical solution to the problem of large-scale facial recognition necessarily exists. Humans are very good at performing facial recognition and thus, in principle, it should be possible to automate this procedure with machine learning. In this sense, the attacks that we evaluate are essentially trying to defer the inevitable (and we show that they will fail at doing that even against oblivious defenders).
>
> In any case, we believe that papers that identify problems in machine learning systems can be of significant importance even if they fail to identify a concrete solution to the problem at this stage.
>
> >  *“however, it can still be considered an arms race with the constraint that the defender is in a better position to win”*
>
> Yes, we agree. This is a question of the semantics of what we mean by an “arms race”.

---

### Decision · Program_Chairs · 2022-01-20

**Decision:**

Accept (Poster)

**Comment:**

This paper reveals that popular data poisoning systems, Fawkes and LowKey, fail to effectively protect user privacy in facial recognition. The methods to defend against poisoning attacks are quite simple---you can either adaptively tune the face recognition models or just wait for more advanced facial recognition systems. Given these “disappointed” findings from the technical solution side, this paper further argues that legislation may be the only viable solution to prevent abuses of facial recognition.

Overall, all the reviewers highly appreciate the comprehensive and rigorous evaluations provided in this paper and enjoy reading it. The biggest concern is raised by the Reviewer 6s7m, given this work fails to discuss/compare to previous works on Facial identity anonymizing and the technical contribution is incremental. During the discussion period, all other reviewers reach a consensus that 1) facial identity anonymizing is not relevant; and 2) this work make enough contributions and is worthy to be heard by the general community; the Reviewer 6s7m still hold the opposite opinion, but is okay for accepting this paper anyway.

In the final version, the authors should include all the clarification provided in the discussion period.